# Caring While Moving: Case Studies on Physical Activity and Dementia Caregiver Well-Being

**DOI:** 10.3390/healthcare13172205

**Published:** 2025-09-03

**Authors:** Jeffrey T. Boon, Cathy A. Maxwell, Diana M. Layne

**Affiliations:** 1College of Nursing, The Ohio State University, Columbus, OH 43210, USA; boon.18@osu.edu; 2College of Nursing, University of Utah, Salt Lake City, UT 84112, USA; cathy.maxwell@nurs.utah.edu; 3College of Nursing, Medical University of South Carolina, Charleston, SC 29425, USA

**Keywords:** family caregivers, dementia, physical activity, case study, mitochondrial fitness, caregiver stress

## Abstract

**Background/Objectives**: Many community-dwelling Americans with Alzheimer’s disease and related dementias rely on family caregivers to help meet their increasing care needs. These caregivers experience increased stress related to caregiving as well as more sedentary lifestyles that lead to the development of chronic non-communicable disease (NCD) and poorer health outcomes. Physical activity interventions for both the caregiver and person living with dementia have the potential to address these issues. We aimed to identify key factors that should be considered in tailoring physical activity interventions for dementia family caregivers. **Methods**: Two case studies of dementia family caregivers are presented that describe the role of physical activity as it relates to their own health and the effects of caregiving. One caregiver was in the early stages of caregiving while the other provided the perspective of a caregiver after the death of the person with dementia. **Results**: While both caregivers participated in physical activity, their case studies reveal opportunities to optimize physical activity interventions to support members of the caregiving dyad: (1) targeting mitochondrial fitness—the ability of mitochondria to maintain energy homeostasis with aging—so participants can mitigate the onset of NCDs, fatigue, and overall health outcomes; (2) promoting the self-care and psychological benefits of physical activity such as improved mood and social connectedness; and (3) improving the accurate perceptions of participants’ health and physical activity with specific measurable markers of physical activity. **Conclusions**: These case studies demonstrate the key features in the development of psychoeducational interventions to promote physical activity in dementia caregiving and will be used in future intervention development.

## 1. Introduction

In 2024, 6.9 million Americans of the age of 65 or older live with dementia due to Alzheimer’s disease [1]. While often underdiagnosed, other types of dementia or mild cognitive impairment increase the number of older adults affected by some degree of cognitive impairment [2,3]. Although some individuals with Alzheimer’s disease or a related dementia reside in some form of institutionalized setting, an estimated 11.5 million unpaid caregivers (family members or other close relationships) provide the majority of the needed care for older adults with dementia [1]. Most community-dwelling adults with dementia do not receive paid care but rather they rely on family members to adapt their own lives to the needs of the person with dementia, handling increasing responsibilities to meet the needs of their loved one [4].

It is well established that persons living with Alzheimer’s disease and related dementias (PLWDs) and their caregivers experience tremendous stress [5]. Amidst the stress and challenges, both PLWDs and caregivers also experience more sedentary lifestyles than their cognitively healthy counterparts [6,7,8]. Physical activity is perhaps the most potent medical intervention for improving health among aging adults (mean age: 54.5) based on a recent large-scale study of over 30,000 subjects [9]. Despite the known benefits of physical activity in promoting healthy aging [10] and reducing stress [11], common barriers to routine physical activity for older adults include fear of injury, disinterest, chronic health problems, and inadequate environmental support [12].

Although clear evidence exists for benefits in the general population, the body of literature on exercise in persons living with dementia (PLWD) and their caregivers shows mixed reports regarding the benefits of exercise interventions. A recent systematic review indicates that exercise interventions improve activities of daily living (ADLs), walking, balance, and visuospatial processing in PLWDs or mild cognitive impairment [13]. Interventions for caregivers and dyadic interventions for both caregivers and PLWDs demonstrate feasibility [14,15], and small to moderate effect sizes for improving physical health [16,17]. Psychoeducational interventions that focus on stress management are both feasible and effective in promoting self-efficacy and reducing stress, depression, and caregiver burden [5,11,15]. Conversely, a 2023 systematic review of randomized controlled trials with older adults living with Alzheimer’s disease showed low evidence for improvement in ADLs and no effect for muscle strength, postural balance, and flexibility [18]. Interventions to address lifestyle change have produced mixed results for numerous reasons including habit change challenges and psychological/cognitive factors (e.g., delayed gratification, underestimating time and effort, etc.). Effective motivation entails addressing the deeply ingrained nature of habits as well as psychological barriers. One intervention, MitoFit, appears to address these barriers through cognitive restructuring—defined as identifying unhelpful and challenging thoughts and replacing these with adaptive thoughts and accurate understanding [19]. This inconsistent evidence highlights the need to harmonize approaches that demonstrate efficacy to improve physical health, reduce stress and caregiver burden, and which are cost effective.

Motivating individuals to engage in physical activity and other lifestyle measures is complex and challenging. Caregiving is an added challenge because many caregivers neglect personal needs to focus on their loved ones. Few studies have viewed caregivers and physical activity through a motivation lens, and the results are mixed. The MitoFit intervention utilizes science communication as an approach to motivate behavior change, and our preliminary work shows promising results [19,20]. Towards this aim, in this case study report, we examine physical activity in two family caregivers of PLWDs to further understand its role in their daily lives. The case study methodology can produce insights that might be missed by other methods and can be a precursor to larger studies; thus, we used it as a first step towards applying MitoFit to caregivers. By seeking to gain in-depth and detailed insight into the complex and multidimensional phenomenon of dementia dyads, we hope to tailor a novel intervention (MitoFit) for these caregiving dyads. As a result, we aim for an intervention that promotes health, wellness, and quality of life through a science communication approach to PLWDs and their family caregivers [19,20].

## 2. Materials and Methods

Two individuals known to the researchers were recruited to complete one-hour semi-structured interviews about their experiences of caregiving for someone with dementia and the role of physical activity in their caregiving experience in June 2023. The interviews were audio recorded and transcribed verbatim. After reviewing the case studies independently, the research team identified key themes in each case through a group consensus process. In consultation with the Institutional Review Board at the Medical University of South Carolina, a formal ethics review was not required with the inclusion of only two participants. Participants provided written permission for the use of their case studies in this publication and were provided the opportunity to review the manuscript prior to submission for publication. Participants also received a USD 25 gift card for the time they gave to share their experiences.

The individuals selected for this case study were chosen as exemplar cases at the beginning and end of the caregiving timeframe to examine both prospective and retrospective experiences of physical activity and caregiving. This sample size was considered sufficient as case study research does not seek to be broadly generalizable but to draw out the important characteristics of a phenomenon from the close examination of the case. The only key inclusion criterion was that the participants attested that they were able to speak to the topics proposed in the interview. For this case study, we chose to forego a formal theoretical framework in favor of an exploratory or inductive approach for early-stage work on the phenomenon which can inform the future selection of a theoretical framework. The priorities for data collection were developed a priori and included in an interview guide (Table 1): (a) description of the caregiver’s own health and physical activity background; (b) description of the caregiving experience for a PLWD including stressors or benefits; (c) the impact of physical activity on caregiving; and d) facilitators/barriers to physical activity. The cases were analyzed by first developing case descriptions according to the priorities identified for data collection followed by the cross-case synthesis completed in consensus discussion to determine the ways in which the cases replicated or contrasted with one another [21].

## 3. Results

The case descriptions below were generated in the analysis process for use by the team to focus on key points in each case study. A side-by-side comparison of the case studies reports the results from the cross-case-synthesis in a format that allows for a comparison between the cases for the reader (Table 2).

### 3.1. Case Study 1

Ms. J is an African American woman in her sixties who recently retired from full-time work in an office setting. She describes her health as “good”, though with “some issues” including hypertension, osteoarthritis (now with a hip replacement), and a cerebral aneurysm. She has been a non-smoker, does not report problems with cholesterol or diabetes, and receives regular healthcare from her primary care provider and ongoing follow-up for the cerebral aneurysm. She describes herself as a person who “watches what she eats and drinks”. Additionally, she endorses trying to stay active but does not have a history of heavy exercise.

Ms. J cares for her 88-year-old mother who lives in another city. Her mother lives independently and has other children nearby who are also involved in her care. Her mother has been described by her doctor as having dementia, although Ms. J does not know what type. Cognitively, she has mild cognitive impairment though she no longer does certain tasks at home like cooking or shopping. She has difficulty making decisions, problem-solving, and remembering how to use some appliances at home. She independently performs some activities of daily living such as toileting and bathing. She also has mild impairment in attention and concentration. Ms. J saw her mother on weekends about every two weeks to help provide care while she was working but plans to see her more and for longer periods now that she is retired.

Ms. J expresses distress as a family caregiver related to seeing changes in her mother. Ms. J feels stress with conflicting emotions over wanting to help her mother continue living in her own home because her mother loves living there. She realizes that the day may come when independent living is no longer an option. Another stressor is wanting to make sure her mother will ask for help when needed because she often expresses that she feels like she is bothering Ms. J. She also realizes that the progression of dementia could also interfere with Ms. J’s ability to ask for help. Ms. J does not identify any other major caregiving-related stressors and feels like she will feel better about caregiving now that she is able to spend more time with her mother.

Ms. J has recently joined a gym for exercise where she goes in the morning to walk on the treadmill, around the track, or to ride a stationary bike. She estimates she goes for about an hour most days. Before joining a gym, she would walk in her neighborhood or on a treadmill at home. She has also started to plant a garden to be more active and keep busy. She says that she participates in physical activity and exercise mostly for health reasons. She identifies risk for heart disease or diabetes as well as wanting to not need a lot of medication. She also recognizes a social benefit to her gym membership, stating that she is an early riser and will be at the gym early with others. Her physical activity provides some stress relief because she can listen to music or a book while on the bike to get her mind off things. She does not identify major barriers to exercise except that she may decide to sleep in some mornings if, for example, it is raining. Facilitators for participating in physical activity are the motivation to be healthy and the social benefit of interacting with others at her gym.

### 3.2. Case Study 2

Mr. M. is a white man in his sixties who is employed in a work-from-home setting and describes himself as being in “decent health”. He has type 2 diabetes, hypertension, and hyperlipidemia all managed with oral medication. He describes himself as having orthopedic problems particularly in one leg because of athletic injuries when he was younger and an accident requiring ankle surgery during college. As a result, he says he has daily chronic pain such that he no longer does heavy lifting but can still do other things such as playing golf for enjoyment. He has regular access to a primary care provider and tries to adhere to their recommendations. He also identifies job-related stress as a contributor to his health problems.

Mr. M was involved in the care of his mother until she died last year at the age of 92. By the end of her life, she had moderate dementia severity with some memory loss, disorientation and dwelling on the past, difficulty with decision making, behavioral and personality changes, and mild anomia. Mr. M acknowledged that these changes resulted in increased stress for him and required him to adapt his coping as a caregiver. She also required someone to accompany her for any activities outside the home as well as significant help with personal care. Mr. M’s mother lived in an assisted-living facility in her own apartment. Mr. M visited nearly daily to see his mother and help with caregiving tasks. Mr. M would be involved in assisting his mother with transfers and ambulation using her walker when he was present. Although he also states he was fortunate to be able to have a paid caregiver help her during the day for ADLs as well as social interaction, the daily responsibilities of overseeing her care contributed to his overall stress and decreased ability to engage in all his leisure activities.

He recognized the challenges of caring for a PLWD, though he also said an unanticipated benefit was that it renewed his appreciation of life. An important motivation for him in caregiving was the desire to take care of his mother “because she had taken such good care of me”. While he expressed the desire to provide care, he recognized that the caregiving experience altered his life by, e.g., reducing time for participation in hobbies like golf and adding additional stress to existing work-related stress. However, Mr. M reported social support from his spouse and an employer who had experienced similar caregiving responsibilities.

In terms of physical activity, Mr. M described himself as having been involved in athletics his whole life. As an adult, he played lunchtime basketball until he was limited by pain. He then took up swimming for exercise as well as occasional walks with his wife. Mr. M says that swimming provided a good stress relief while providing care for his mother because it provided time to not be talking to anyone or doing something for someone else. He identified fatigue related to caregiving as a major barrier to physical activity, but he also was able to identify facilitators such as having a swimming pool a few miles from where his mother was living so he could get his exercise before or after helping with his mother’s care. Mr. M said that he recognized the importance of his physical activity to caregiving because of the importance of self-care in caregiving.

## 4. Discussion

Three key issues are imbedded in the two case studies that directly relate to physical activity in family caregivers for people living with dementia. The development of chronic disease, which is a common but not a normal part of aging, was seen in both family caregivers. These kinds of health conditions can impact the family caregivers’ well-being and be prevented, delayed, or mitigated by attention to mitochondrial fitness. The family caregivers also emphasize the benefits of physical activity for their psychological well-being and self-care, but there are further opportunities to be explored such as dyadic physical activity that may enhance these benefits. Finally, the case studies demonstrate that it is important to pay attention to caregivers’ perceptions of their own health and physical activity so that interventions can optimize the benefits to caregivers by addressing these perceptions.

The insights obtained from the case studies about the caregivers’ perceptions of themselves and physical activity are particularly fitting for an intervention model like MitoFit. Despite having self-perceptions of being in good health, both have chronic NCDs that could have been avoided through a better understanding of the science behind physical activity [19]. That understanding is a key aspect of MitoFit. Furthermore, the participants’ recognition of the psychological benefits of physical activity would potentially match with the practicality and ease of applying the MitoFit approach [20].

While there are numerous insights that can be gleaned from the case study approach, it does not have the same goals or reach of a large study in terms of generalizability or being able to account for interpersonal differences such as diverse cultural backgrounds, different care settings, or the different stages of dementia. However, the key findings are broad and should likely be considered in adaptation to any of the unique circumstances for potential participants in a physical activity intervention.

### 4.1. Chronic Non-Communicable Disease and Mitochondrial Fitness

The two caregivers presented in this case study had several commonalities that give key insights into the importance of physical activity in their lives. In terms of physical health, both had NCDs such as hypertension, hyperlipidemia, and type 2 diabetes mellitus that are common in adults in the United States. Family caregivers, especially those experiencing high levels of caregiver strain like Mr. M, are at increased risk for poorer self-care related to hypertension and diabetes, including reduced physical activity [22,23]. Physical activity has been repeatedly recognized as a key factor influencing NCDs and their impact on society as a whole [24,25].

Likewise, both caregivers expressed some degree of fatigue affecting their daily lives, even to an extent that it was sometimes a barrier to physical activity for exercise. Fatigue in dementia family caregivers is especially associated with high-intensity caregiving and incidentally, also with lower levels of physical activity by the caregivers [26]. While fatigue has poorly understood etiologies, it has been linked at the subcellular level to disruption in bioenergetic homeostasis, particularly in the form of mitochondrial dysfunction [27]. While mitochondrial processes are linked to fatigue, evidence suggests that disrupted carnitine levels—critical for fatty acid transport and oxidation—may represent a key mechanistic link between mitochondrial energy homeostasis and fatigue [28].

In addition to its link to fatigue, mitochondrial fitness (the capacity of mitochondria to maintain energy homeostasis) is also associated with the development of NCDs [29,30]. Physical activity is linked to improved mitochondrial function, and importantly, represents a mechanism for the potential reversal of decline in adults [31,32,33,34,35]. Because family caregivers have been identified at particular risk for NCDs and fatigue due to the extra stressors of caregiving, physical activity targeting an underlying cellular mechanism, mitochondrial function, holds potential to have benefit for family caregivers.

Indeed, pioneering work by Picard and colleagues shows that chronic caregiving stress and daily mood influence mitochondrial function, and this may be key to understanding the connection between psychological stress and poor health outcomes [36,37]. Better mitochondrial respiratory capacity in dementia spousal caregivers has also been associated with better health [38]. This suggests the special importance of mitochondrial fitness for family caregivers.

Physical activity interventions for older adults have been developed specifically to target mitochondrial fitness. One such program is MitoFit, a science communication intervention to promote behavior change through education about the concept of mitochondrial fitness [19,20]. This program was developed in response to feedback from another healthy aging program in which older adults expressed interest in more information on the concept of energetics and mitigating cellular aging [39]. MitoFit targets mitochondrial fitness in two ways. In the first half of the program, the participants viewed educational videos about the role of mitochondria in health and disease and maintaining bioenergetic homeostasis by participating in physical activity targeting mitochondrial function.

In the second half of the program, the participants learned how to implement and track a physical activity program consisting of zone 2 aerobic activity [40] and strength exercises. Teaching individual heart rate zones and how they contribute to mitochondrial fitness is a key element of the MitoFit intervention. Heart rate zones 1 and 2 are zones in which mitochondria utilize fatty acids—the preferred fuel (substrate) for optimal mitochondrial function. Zones 1 and 2 are not difficult and allow a person to carry on a conversation while walking. As one’s endurance increases, the resting heart rate decreases and indirect maximal oxygen consumption (VO_2_Max) increases, enabling one to experience the benefits firsthand and know that they are achieving mitochondrial fitness. Over time (six months), the principles of habit formation reinforce sustainability. These are science communication principles taught via the intervention. Programs such as this have the potential to meet family caregivers’ physical activity needs and should be tested and adapted as needed for this population.

The literature reports abundant barriers to physical activity, particularly among older adults [41]. An attraction of MitoFit is its practicality, the perception of less burden, and the ability to carry it out in one’s own home if needed. For example, MitoFit focuses on walking for 20–30 min with heart rate zones 1 and 2. This can be done in a neighborhood, an inside space, or walking in place. When individuals perceive that physical activity does not have to be strenuous, they are more likely to engage in physical activity [42]. Barriers identified by caregivers of PLWD differ from those of non-caregivers. The MitoFit pilot study identified several themes and codes that reflect the unique barriers of caregivers [19]. Table 3 compares the barriers to physical activity experienced by caregivers with themes and codes from the MitoFit pilot study that show how the physical activity intervention can be targeted towards caregivers’ perceived barriers.

### 4.2. Self-Care and Psychological Benefit

Both caregivers in the case studies indicated the importance of physical activity as a form of self-care. While Mrs. J indicated that her primary reason for physical activity is health, she also indicated the stress-relief and social benefits of physical activity and going to her gym. Mr. M. indicated less emphasis on a social benefit and focused on the way in which his swimming allowed him to escape from stressors for a period while also doing something for his health.

Indeed, studies have demonstrated benefits from physical activity interventions for caregivers on their mental health [43,44]. Loneliness has been found to have a high prevalence among family caregivers of PLWD [45]. Loneliness can be viewed through a lens of perceived social isolation which can be a stressor for many individuals, particularly older adults. Chronic loneliness is a public health issue that contributes to mental health disorders and poor metabolic health [46]. Physiologic mechanisms and pathways entail loneliness as a chronic stressor that leads to the dysregulation of the hypothalamic–pituitary–adrenal axis with immune and metabolic consequences that manifest in chronic disease. Mitochondrial dysfunction is a foundational cause of these processes [46] that can be mitigated or attenuated through physical activity [47]. In a review of the interventions to reduce loneliness in caregivers, physical activity was not among the interventions reviewed [48]. However, even mild physical activity has been found to moderate the impact of loneliness on depressive symptoms in older adults [49]. Other evidence suggests the that the benefits of group physical activity may also improve social isolation, loneliness, and social connectedness [50,51]. This suggests that physical activity may have some benefit for family caregivers’ experience of loneliness and affective symptoms. Overall, a mitochondrial fitness approach contributes to both physical (increased endurance and strength) and cognitive (decreased anxiety/depression, increased hippocampal function, problem solving) benefits [52].

Both caregivers described here participated in exercise separately from the PLWD care recipient. While matching physical activity capabilities may provide some challenges in the caregiver–care recipient relationship, there is evidence of the potential benefits for dyadic physical activity. In particular, dyadic physical activity interventions have been shown to have positive effects for family caregivers including caregiver burden, some mental health outcomes, caregiver quality of life, and relationship quality [14,16]. The value of dyadic physical activity has been more fully studied in other settings such as romantic partnerships with growing evidence to support a dyadic version of the theory of planned behavior [53]. This provides the hope for future dyadic, theory-driven interventions for dyadic physical activity in caregiving relationships and is an important area for future study.

### 4.3. Perceptions of Health and Physical Activity

Interestingly, both individuals presented in these case studies use positive language to describe their health. Indeed, both use language that is mostly positive—Ms. J describing her health as “good” and Mr. M describing his as “decent—” followed a qualifier such as having “some issues” for Ms. J or description of a history of problems such as Mr. M’s chronic pain history. On further inquiry, both described having NCDs that could be prevented or mitigated with the right types of physical activity. These case studies did not explore the reasons for the contrast in the participants’ own perceptions of their health and their medical conditions. However, factors in self-perception such as optimism bias support the importance of cognitive approaches to health behavior changes [54,55]. Addressing optimism bias can be a key factor in intervention development. Indeed, the science communication component of MitoFit clearly identifies chronic NCDs that are not representative of normal health for older adults and can be prevented or mitigated by physical activity. Cognitive restructuring in science communication approaches presents new information to align previously held beliefs with ones more congruent with reality [19].

Likewise, the case studies illustrate the importance of clear communication about physical activity for caregivers at their life stage. Mr. M compared his current activity with a life history of participating in athletics which was limited by chronic pain development that led to his current exercise regimen. While this seems a logical pathway in one sense, it does not indicate knowledge of physical activity goals for his age and health needs such as 150 min of moderate-intensity aerobic activity and resistance exercise [56]. While swimming is an activity capable of achieving the intensity desired, Mr. M’s responses do not demonstrate any measurement of his exercise intensity, such as heart rate, that could be used as a proxy for VO_2_Max.

Ms. J and Mr. M also did not provide specific measures of frequency and consistency for their physical activity. Ms. J states that she exercises for about an hour on most days, but “most days” is relatively non-specific. The value of understanding and accurately measuring physical activity is brought to light when considering that people tend to overestimate their own levels of physical activity [57]. Even strong self-report measures of physical activity have long been recognized to have challenges in providing an accurate assessment of physical activity [58]. Interventions like MitoFit can be adapted to address these concerns. Any type of aerobic (zones 1 and 2) activity is beneficial towards mitochondrial fitness, as evidenced by reductions in inflammatory markers [59,60]. This is useful for addressing the needs of participants with different health limitations or other circumstances. The MitoFit videos discuss a variety of aerobic activities that individuals can engage in, including walking, yoga, gardening, dancing, and others. This is applicable to people with and without dementia and can be done by the caregiver with or without dyadic participation. To promote reaching the necessary targets (heart rate), integrating wearable technology is a possibility to provide immediate feedback, though there remains a need to validate wearable technologies [61].

The importance of being able to plan physical activity that meets appropriate measures of frequency and intensity is crucial for caregiver physical activity interventions. Interventions should provide appropriate education on these principles to increase the chances that participants will adhere in ways that help them achieve beneficial physical activity. For example, findings from a recent feasibility study of a nurse-led early palliative care intervention-A Program of SUPPORT-D^TM^—highlighted that dementia caregivers specifically requested more guidance on incorporating physical activity into their routines [62]. SUPPORT-D^TM^, which offers anticipatory guidance for persons living with dementia and their caregivers, was found to be both feasible and acceptable, and participants specifically endorsed the inclusion of physical activity education as a valuable enhancement for future iterations of the program. Evidence suggests that caregivers of PLWD were motivated to participate in group exercise activities with others at a similar cognitive/caregiver stage [63].

Complementing this need, science communication models such as MitoFit provide a promising framework for the delivery of this type of education. Mitofit not only explains the role of mitochondria in aging and physical activity but also on markers of effective exercise such as heart rate/VO_2_Max and frequency/duration of exercise. The hope is that this educational approach will better align with people’s perceived and actual levels of activity.

## 5. Conclusions

Physical activity has been recognized to be a key part of healthy aging—preventing or addressing non-communicable diseases while also providing a psychological benefit. For family caregivers who experience additional stressors related to their caregiving activities, the benefits of physical activity may be even more crucial. Physical activity interventions that are targeted to adults’ healthy aging needs such as mitochondrial fitness as well as the specific needs for self-care in family caregiving should be developed to optimize health and well-being for family caregivers.

## Figures and Tables

**Table 1 healthcare-13-02205-t001:** Guiding questions for the semi-structured interview with the family caregivers of a person living with dementia.

Please tell me about yourself (include health history questions).
How would you describe you overall health?
Please tell me about the care recipient (information about the person, type of dementia, etc).
Please tell me about your caregiving experience.
What is your experience with physical activity?
Do you believe physical activity helps to manage stress? Why or why not?
What makes it difficult to participate in physical activity?
What makes it easier to participate in physical activity?

**Table 2 healthcare-13-02205-t002:** Comparison of the key aspects of case studies of family caregivers of people living with dementia.

	Ms. J	Mr. M
Age *	Sixties	Sixties
Relationship to care recipient	Daughter	Son
Employment status	Retired	Work-from-home full time
Self-perception of health status	“Good” with “some issues”	“Decent health”
Health conditions	Hypertension, osteoarthritis (hip replacement), cerebral aneurysm	Type 2 diabetes mellitus, hypertension, hyperlipidemia, chronic pain/joint injuries
Physical activity background	Tries to stay active but no history of heavy exercise	Athletics for much of his life then took up swimming for exercise
Physical activity while caregiving	Exercise at a gym (treadmill, walking on track, stationary bike). Previously on a treadmill at home, walking in neighborhood, gardening	Mostly swimming for exercise, sometimes walks with his spouse
Frequency or other measures of physical activity	About an hour most days	Not clearly quantified by participant
Impact of physical activity on caregiving	Social benefit of gym membership. Stress relief.	Stress relief. Identifies value of being alone.
Facilitators of physical activity	Motivation is for health reasons, especially not wanting to need a lot of medication.	Accessibility of swimming pool. Personal recognition of importance of self-care in caregiving.
Barriers to physical activity	May decide to sleep in. Weather (raining in morning).	Fatigue

* Some characteristics are not described with exact numbers (e.g., age) unless necessary for the case study to preserve participant privacy.

**Table 3 healthcare-13-02205-t003:** Caregiver barriers to physical activity aligned with themes and codes from the MitoFit pilot intervention study.

Caregiver Barriers to Physical Activity	Themes/Codes (Perceptions) and Definitions Related to MitoFit Identified by Adults Aged 50+ That Address Caregiver Barriers [19]
**Limited time**	**Making a plan**—Recognition that mitochondrial fitness requires a plan that one perceives as doable **Accessible**—Realization that MitoFit is practical and attainable **Finding what works**—Appreciation for the ability to choose activities that one perceives as individually achievable IMPLICATION: Increased perception of feasibility
**Lack of support**	**Sense of Agency**—Subjective feeling of ability to achieve a goal **Removes moral judgement**- Sense that MitoFit content reduces feeling of shame or guilt by replacing with awareness of scientific foundations IMPLICATION: Enhancement of self-reliance
**Scheduling constraints**	**Options and Variety**—Recognition that MitoFit offers variety and options for PA **Simple**—Realization that PA does not have to be hard and can be achieved in one’s own home IMPLICATION: Diminishment of roadblocks to PA
**Health conditions/fatigue**	Positive Emotions—Pleasant state of mind **Enjoyable** **Good** **Helpful** **Makes sense** **Nice** IMPLICATION: Positive perceptions contribute to the likelihood of engaging in physical activity
**Lack of motivation**	**Actionable**—Perception of practical value **Affirming**—Sense of support and encouragement **Steady state**—Perceptions of stability over time Leaves you wanting more—Sense of anticipation to learn more and to engage in PA IMPLICATION: Perceptions that minimize obstacles that prevent initiation and sustainability of PA

## Data Availability

Data are not publicly available due to the sensitive nature of the data collection.

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
