# Peer review of "Caring While Moving: Case Studies on Physical Activity and Dementia Caregiver Well-Being"

_healthcare, 2025, doi:10.3390/healthcare13172205_

Round 1

Reviewer 1 Report

Comments and Suggestions for Authors

It was a pleasure to review this manuscript, which addresses an important and underexplored topic — the role of physical activity in dementia family caregivers. The manuscript is generally well written, coherent, and supported by a clear rationale. The use of two case studies offers valuable qualitative insight into different stages of the caregiving trajectory and their relationship with physical activity. The focus on mitochondrial fitness as a conceptual framework is innovative and aligns with emerging evidence on cellular health and ageing.

Nonetheless, several aspects could be improved to strengthen the clarity, theoretical grounding, and methodological detail:

Introduction
L32–71 — While the background is comprehensive, it would benefit from a clearer articulation of the knowledge gap the study seeks to address. After summarising the mixed evidence from existing literature, explicitly state why a case study approach focusing on mitochondrial fitness offers new insights. Also, better explain how this approach fits into the current landscape of dementia caregiver interventions.

L52–64 — The review of existing evidence is strong, but there is an opportunity to highlight more explicitly why previous interventions have produced mixed results, including possible methodological and contextual factors.

Methods
L72–90 — Please justify why a formal theoretical framework for data collection/analysis was not applied, and discuss how the absence of such a framework might influence interpretation of results.

L73–83 — Clarify whether the interview guide was piloted or adapted from validated tools. If developed from scratch, please justify and include examples of guiding questions.

L84–90 — Explicitly state why two participants were considered sufficient for this case report and how they were selected. Include any inclusion/exclusion criteria applied.

Case descriptions

L91–132 & L133–173 — Consider reorganizing the case studies to present key socio-demographic and health background information in a structured format before the narrative, to improve readability and comparison between cases.

Discussion

L175–185 — The three identified key issues are relevant, but the link between the case findings and the proposed intervention model (MitoFit) could be more explicitly integrated here rather than later in section 4.1.

L186–229 — The discussion on mitochondrial fitness is strong but could benefit from a more critical perspective on potential barriers to implementing such interventions in real-world caregiving contexts (e.g., time constraints, accessibility of facilities, financial costs).

L230–256 (Self-care and psychological benefit) — Expand the discussion on how physical activity can address loneliness in caregivers, possibly citing additional literature from non-dementia caregiving contexts for a broader perspective.

L257–284 (Perceptions of health) — Consider linking the optimism bias discussion to practical implications for designing educational components in interventions, beyond simply stating the phenomenon.

L275–299 — When discussing physical activity targets and measures, provide more specific guidance/examples relevant to caregivers with health limitations, ensuring inclusivity.

Throughout — While the English is clear, some sentences are long and could be split for better readability. For example, in L68–71, consider breaking into two sentences to enhance clarity.

Overall, this is a valuable and timely contribution. Addressing the points above will further enhance its methodological transparency and strengthen its relevance to clinical and public health practice.

Author Response

Thank you for insightful feedback and the opportunity to revise our manuscript. The attached reviewer table responds to each comment individually by reviewer.

Reviewer 2 Report

Comments and Suggestions for Authors

This case report provides valuable insights into the role of physical activity in supporting the health and well-being of dementia family caregivers. The use of two contrasting cases—one in the early stages of caregiving and one in the post-caregiving stage—offers a nuanced perspective on how caregiving responsibilities intersect with exercise habits, health perceptions, and barriers to self-care. The integration of mitochondrial fitness as a physiological framework is novel and adds mechanistic depth to the discussion, linking cellular energetics to caregiver health outcomes.

The manuscript is well-organized, clearly written, and appropriately contextualized within the existing literature. It makes a meaningful contribution to the field by highlighting intervention design considerations that go beyond general exercise promotion, including the importance of targeting mitochondrial health, fostering psychosocial benefits, and improving accurate self-perception of physical activity.

To further strengthen the manuscript, the following suggestions are offered:

  1. Expand methodological transparency – While the rationale for using two exemplar cases is clear, providing more detail on participant selection criteria and interview procedures (e.g., thematic coding process, researcher positionality) would enhance the trustworthiness of the findings.
  2. Clarify mitochondrial fitness application – The concept is central to the discussion, yet some readers may not be familiar with “zone 2 aerobic activity” or specific physiological markers. A brief, practical explanation and example of measurable indicators (e.g., heart rate ranges, VOâ‚‚Max targets) would improve accessibility for multidisciplinary audiences.
  3. Address generalizability – The two cases provide depth but are limited in scope. A short reflection on how findings might translate to caregivers from diverse cultural backgrounds, care settings, or disease stages would broaden applicability.
  4. Integrate objective measurement perspectives – As both participants relied on self-reported activity, it may be helpful to briefly discuss the potential role of wearable technology or accelerometers in future studies to improve accuracy and intervention tailoring.
  5. Consider visual synthesis – A summary table outlining caregiver characteristics, key barriers/facilitators, and implications for intervention design could increase clarity and aid practitioners in quickly extracting actionable points.

Overall, this is a compelling and forward-looking case report that bridges physiology, psychology, and caregiving research. With minor refinements to methodological detail, practical application, and visual presentation, the paper has strong potential to inform both future research and the development of tailored physical activity interventions for dementia caregivers.

Author Response

Thank you for your fvaluable feedback. Please see the attached reviewer table which responds to each item individually by reviewer.

Reviewer 3 Report

Comments and Suggestions for Authors

The manuscript needs the following clarifications:

  1. The study is related to tailoring physical activity interventions for dementia family caregivers, but that is not reflected in the title of the manuscript. Please include the term dementia in the title itself
  2. Methods: Better to mention the study time period i.e. month, year.
  3. PLWD – please include the full form when using for the first time in the manuscript
  4. Line 230 – 256: Most of the statements are related to physical activity, which has already been described in the previous paragraph. Thus, it would be better to include specific benefits from self-care and psychological benefits.
  5. It would have been better to detail a little bit about specific types of physical activity that the respondents think would be beneficial for people with dementia.

Author Response

Thank you for the opportunity to revise our manuscript and the valuable feedback you provided to strengthen the manuscript. Please see attached for a detailed response to each item of reviewer feedback organized by reviewer.
